# *Pseudomonas aeruginosa* in the Frontline of the Greatest Challenge of Biofilm Infection—Its Tolerance to Antibiotics

**DOI:** 10.3390/microorganisms12112115

**Published:** 2024-10-22

**Authors:** Niels Høiby, Claus Moser, Oana Ciofu

**Affiliations:** 1Institute of Immunology and Microbiology, Costerton Biofilm Center, Faculty of Health Science, University of Copenhagen, DK-2200 Copenhagen, Denmark; moser@dadlnet.dk (C.M.); ociofu@sund.ku.dk (O.C.); 2Department of Clinical Microbiology, Rigshospitalet, University of Copenhagen, DK-2100 Copenhagen, Denmark

**Keywords:** *Pseudomonas aeruginosa*, biofilm, chronic infection, antibiotic tolerance

## Abstract

*P. aeruginosa* biofilms are aggregates of bacteria surrounded by a self-produced matrix which binds to some antibiotics such as aminoglycosides. *P. aeruginosa* biofilms are tolerant to antibiotics. The treatment of biofilm infections leads to a recurrence of symptoms after finishing antibiotic treatment, although the initial clinical response to the treatment is frequently favorable. There is a concentration gradient of oxygen and nutrients from the surface to the center of biofilms. Surface-located bacteria are multiplying and metabolizing, whereas deeper located bacteria are dormant and tolerant to most antibiotics. Colistin kills dormant bacteria, and combination therapy with colistin and antibiotics which kills multiplying bacteria is efficient in vitro. Some antibiotics such as imipenem induce additional production of the biofilm matrix and of chromosomal beta-lactamase in biofilms. Biofilms present a third Pharmacokinetic/Pharmacodynamic (PK/PD) micro-compartment (first: blood, second: tissue, third: biofilm) which must be taken into consideration when calculations try to predict the antibiotic concentrations in biofilms and thereby the probability of target attainment (PTA) for killing the biofilm. Treating biofilms with hyperbaric oxygen to wake up the dormant cells, destruction of the biofilm matrix, and the use of bacteriophage therapy in combination with antibiotics are promising possibilities which have shown proof of concept in in vitro experiments and in animal experiments.

## 1. Introduction

*Pseudomonas aeruginosa* is one of the ESKAPE bacterial pathogens which are identified as critical multidrug-resistant bacteria for which new effective therapy is needed. These include *Enterococcus faecium*, *Staphylococcus aureus*, *Klebsiella pneumoniae*, *Acinetobacter baumannii*, *Pseudomonas aeruginosa* and *Enterobacter species* [1]. These pathogens thrive in health care environments and they have intrinsic and acquired resistance mechanisms toward antibiotics that contribute to the problem of resistant infections [1]. However, the resistant infections they cause are acute infections resulting from planktonically growing bacteria, e.g., *P. aeruginosa*, which may be resistant to carbapenems or to colistin, whereas their ability to cause chronic infections due to biofilm growth is generally not described or mentioned [1]. Biofilm infections are tolerant to all antibiotics and to the defense mechanisms of patients and induce chronic inflammation which destroys the surrounding tissue (collateral damage). Biofilm infections can also be the focus for the spread of infections to, e.g., the blood stream [2]. Biofilm infections may be found in all anatomical locations of patients (Figure 1). The most comprehensive studies on human biofilm infection, however, have been carried out on chronic *P. aeruginosa* infections in the respiratory tract of cystic fibrosis (CF) patients, which is the topic of this review [3,4].

The present review, therefore, describes the mechanisms behind the tolerance of *P. aeruginosa* biofilms to antibiotics and discusses the possibility of circumventing this phenomenon [3] to eliminate biofilm infections and cure patients.

## 2. The Biofilm Growth of Bacteria

Bacterial biofilms are aggregates of bacteria that are embedded in an extracellular polymeric matrix [3]. Most bacterial species grow as biofilms on submerged surfaces in nature such as on stones in fresh water or salt water [5,6] where the biofilm matrix and size protect them from predators such as amoebae and bacteriophages [7]. The natural habitat of *P. aeruginosa* is water, lakes, rivers, soil and sediments [8]. *P. aeruginosa* is extremely versatile, being able to survive in a broad range of habitats, where most of the microorganism’s cells grow as biofilms attached to surfaces which become slimy or grow as suspended aggregates. In health care settings, *P. aeruginosa* biofilms can be found in sinks, showers, respirator equipment and the water of dental water-cooled drilling equipment [9,10,11]. Aggregates of *P. aeruginosa* become tolerant to antibiotics such as tobramycin when the number of aggregating cells is ≥64, and the tolerance continues to increase with the increasing size of the aggregates [12,13]. When *P. aeruginosa* cause biofilm infections in humans, the aggregates are small (4–200 µm) and difficult to detect. A specific problem is that neither PCR nor culture can distinguish between planktonic and biofilm-growing bacteria, and microscopic detection of aggregates of *P. aeruginosa* is therefore the preferred diagnostic method, which can be species-specific if the fluorescent in situ hybridization (FISH) technique is employed [3,14]. However, in cases of chronic biofilm lung infections in CF patients and in primary cilia dyskinesia patients, culture will in most cases show the growth of mucoid alginate-producing *P. aeruginosa*, and at the same time, the patients will produce high levels of antibodies against *P. aeruginosa*, which is therefore used to diagnose chronic biofilm infections in such patients [3].

There is a gradient of metabolic activity from the surface to the center of *P. aeruginosa* biofilms; generally, the center of biofilms is dormant due to a lack of oxygen and nutrients, and the number of ribosomes in the bacterial cells, which correlates to the growth rate, is low both in vitro and in CF patients [15,16]. The growth of *P. aeruginosa* biofilms in the lungs of CF patients is therefore very slow [17]. This is probably also the reason why intensive antibiotic treatment of *P. aeruginosa* biofilms in CF patients does not eliminate the biofilms but reduces their size—probably by killing their metabolic active surface—and thereby also reduces the inflammation in the lungs (Figure 2) [18].

## 3. Tolerance Mechanisms of *P. aeruginosa* Biofilms

### 3.1. The Biofilm Matrix

Generally, the Pharmacokinetics/Pharmacodynamics (PK/PD) of antibiotics on biofilm-growing *P. aeruginosa* follows the same rules as the PK/PD on planktonically growing bacteria, e.g., time-dependent, concentration-dependent or dose-dependent killing [13,19]. However, some antibiotics such as tobramycin are bound to the alginate matrix of *P. aeruginosa* biofilms [20] and to extracellular DNA in the matrix [21], and the activity of such antibiotics against biofilm-growing *P. aeruginosa* is therefore delayed and reduced. In general, biofilms formed by the mucoid phenotype are more tolerant than biofilms of non-mucoid phenotypes [13]. Furthermore, some antibiotics such as imipenem induce increased ampC beta-lactamase production, increased alginate production (the biofilm matrix) and an increase in the biofilm volume of *P. aeruginosa* biofilms in vitro [22]. From a PK point of view, therefore, it is necessary to describe *P. aeruginosa* biofilms as a third micro-compartment (Figure 3) and take that into consideration in the calculation of the probability of target attainment (PTA) of antibiotics for biofilm treatment [23]. That is probably part of the explanation why the eradication of *P. aeruginosa* biofilms by antibiotics is so difficult in clinical practice.

### 3.2. Antibiotics and Anaerobic Condition in Biofilms

Most of the oxygen around *P. aeruginosa* biofilms in CF sputum is consumed by the surrounding polymorphonuclear leucocytes and to a smaller degree by the metabolism of *P. aeruginosa* [24]. There are, therefore, anaerobic conditions around *P. aeruginosa* biofilms in the sputum inside the conductive airways of CF patients, but *P. aeruginosa* can grow anaerobically—although slowly—because there is enough nitrate in the sputum to function as electron acceptors [25,26]. Most antibiotics (e.g., beta-lactams, aminoglycosides, fluoroquinolones) which are active against *P. aeruginosa* require metabolic activity to inhibit (bacteriostatic) or kill (bactericidal) the bacteria. Therefore, slow-growing or dormant bacteria will either require much longer exposure times to such antibiotics or, in the case of non-growing bacteria, these antibiotics fail to be bactericidal [27]. The intra-bacterial concentration of reactive oxygen radicals (ROSs) produced during bacterial metabolism is increased during exposure to most antibiotics which interfere with their metabolism, and ROSs contribute to the killing of the bacteria by such antibiotics or induce mutations if the bacteria survive [28,29,30,31]. However, ROSs are not formed by dormant bacteria.

The antimicrobial action of colistin, however, is independent of ROSs [32]. Nevertheless, an inducible adaptive tolerance phenomenon toward colistin has been described in *P. aeruginosa*. LPS is produced intracellularly and transported from the cytoplasmic membrane to the outer membrane [33]. Colistin kills *P. aeruginosa* by targeting lipopolysaccharide (LPS) in both the outer membrane and the cytoplasmic membrane, leading to disruption of the cell envelope and bacterial lysis [33]. Colistin with its positively charged L-2,4-diaminobutyric acid binds to the negatively charged LPS phosphates in the outer membrane, displacing Ca^2+^ and Mg^2+^, which bridges LPS molecules. This disrupts the structural integrity of the membrane and thereby self-promotes other colistin molecules to pass through to the cytoplasmic membrane where they bind to the intracellularly produced LPS awaiting to be transported to the outer membrane. Colistin thereby lyses the bacterial cell [33,34].

The mutational resistance mechanisms against colistin involve the alteration of the outer membrane to become less permeable, e.g., through the addition of 4-amino-L-arabinose to LPS which results in reduced negative charges and interferes with the electrostatic interaction with the positively charged colistin molecule [34,35,36,37]. *P. aeruginosa* is known for its antibiotic tolerance due to biofilm growth, but *P. aeruginosa* has another inducible tolerance mechanism which functions similarly to the mutational resistance described above and which also interferes with the action of cationic peptide antibiotics like colistin and other polymyxins [34]. This mechanism was already detected in 1975 [38,39]. Later, it was shown that the polymyxin tolerance of *P. aeruginosa* is due to a complicated two-component regulatory system which also modifies LPS in the outer membrane by adding N4-aminoarabinose to the lipid-A-phosphates of LPS [40]. However, there is a lag period before the tolerance phenomenon occurs since the modified LPS has to be produced intracellularly and inserted in the cytoplasmic membrane and then transported to the outer membrane where it replaces the normal LPS in newly produced cells [33].

It has been shown that if a *P. aeruginosa* biofilm in an in vitro flow-cell experiment is treated with ciprofloxacin, only the metabolic active surface of the biofilm is killed [41]. If, on the other hand, the biofilm is treated with colistin, only the dormant center of the biofilm is killed, whereas the metabolic active surface becomes resistant due to the inducible tolerance mechanism described above [41]. If the biofilm is treated with a combination of colistin and ciprofloxacin (or another anti-pseudomonas antibiotic such as tobramycin or tetracycline), then the whole biofilm is killed [41]. The same phenomenon has been reported with tobramycin and colistin (Figure 4). This synergistic combination has been shown to have better clinical effects in a clinical pilot project in CF patients with chronic *P. aeruginosa* biofilm infection who were treated with this combination for 28 days, where the number of *P. aeruginosa* colony forming units (cfu) decreased by 2.5 log10 (*p* < 0.027) [42]. Furthermore, the combination of inhaled colistin and oral ciprofloxacin was also shown to be able to eradicate intermittent *P. aeruginosa* colonization of the lungs and thereby prevent chronic biofilm *P. aeruginosa* lung infection in CF patients [43].

Colistin combined with rifampicin against *P. aeruginosa* biofilms was likewise found to act synergistically and to kill the biofilms of colistin-resistant strains in vitro, probably because colistin enhances the penetration of rifampicin through the outer membrane of *P. aeruginosa* [44].

## 4. The Way Forward—Circumvention of the Antibiotic Tolerance of *P. aeruginosa* Biofilms

Effective antibiotic therapy for planktonic infections caused by, e.g., *P. aeruginosa* and other ESKAPE pathogens is based on the results of in vitro susceptibility testing of individual antibiotics and combinations of antibiotics. However, the established methods of susceptibility testing rely on planktonically growing bacteria and not on established biofilms of the same bacteria. The disk diffusion method, however, illustrates the tolerance of biofilm growth to the diffusing antibiotics since formation of the edge of the inhibition zone is the transition of the bacteria from the planktonic to biofilm mode of growth [12]. Biofilms adhering to microtiter plates have been used to measure their antibiotic susceptibility (Minimum Biofilm Eradication Concentration) but the results obtained in vitro did not correlate to the results of animal experiments or to the clinical success of antibiotic treatment [13,19,45,46]. The best way to study the action of antibiotics on *P. aeruginosa* biofilms is probably confocal laser scanning microscopy (CLSM) of biofilm formation in flow cells, since the action of antibiotics on both the surface and the center of the biofilm can be visualized, but this method is only suitable for research purposes (Figure 4) [41,42]. At present, the way forward is animal experiments based on promising results from CLSM of biofilms exposed to antibiotics and eventually followed by clinical trials (Figure 4) [42].

Realizing that the dormant interior of biofilm growth is due to a lack of oxygen, in vitro experiments (Figure 5) and animal experiments have been carried out employing hyperbar oxygen atmospheric conditions with promising results and may be a way forward which deserves further study [46,47].

### 4.1. Topical Antibiotic Treatment of Biofilm Infections

The PK/PD problem of the third micro-compartment of bacterial biofilms is difficult to solve due to side effects of antibiotic therapy if too high dosages are used (Figure 6 and Figure 7) [13,49]. However, topical dosing of antibiotics, e.g., by inhalation, may circumvent the problem of most side effects since inhaled antibiotics are virtually not absorbed into the blood, and this is, therefore, an established way of therapy which improves the clinical condition of patients, although it does not eradicate the biofilm infection [48,50,51,52].

However, in the case of *P. aeruginosa* biofilms in the paranasal sinuses of CF patients, treatment with a combination of endoscopic sinus surgery and local application of 5 mL autologous platelet-rich fibrin sealant containing high concentrations of colistinmethate sodium (2.5 mL = 625,000 IU) combined with ciprofloxacin (2.5 mL = 7.5 mg) completely eradicated the *P. aeruginosa* biofilms because the fibrin sealant retained the antibiotics for 1–2 weeks in the sinuses [53], whereas nasal irrigation with antibiotics cannot reach all sinuses and the antibiotics disappear rapidly from the sinuses [54].

### 4.2. Bacteriophage Therapy

Bacteriophage therapy is still in the experimental stage, but current in vitro experiments point toward combinations of bacteriophages and antibiotics like ciprofloxacin to avoid the development of resistance to the phages (Figure 8) [55,56]. However, we have shown that the sputum from 16 CF patients with chronic *P. aeruginosa* biofilm lung infection contained a high number of free bacteriophages belonging to the *Myoviridae*, *Siphoviridae* or *Podoviridae* families. They are all tailed phages which are known to be temperate and thus mediate the transduction and conversion of *P. aeruginosa*. These phages were able to lyse several different clinical *P. aeruginosa* strains in vitro [57]. The therapeutic possibility of bacteriophages in CF patients may therefore be difficult or not possible to realize since the patients’ own *P. aeruginosa* strains already contain bacteriophages which have no therapeutic effects [57]. An in vitro study of the addition of a cocktail of 10 different bacteriophages to the sputum of CF patients with chronic *P. aeruginosa* lung infection showed a reduction in the number of bacteria in the sputum but no clearance of *P. aeruginosa* [58]. The rationale of bacteriophage therapy in CF is clear and a few case reports were published a few years ago, but no breakthrough has been reported [59]. A recent review summarized the problem of the viscid sputum for bacteriophage therapy in CF patients [60].

### 4.3. Destruction of the Biofilm Matrix

Destruction of the alginate matrix of *P. aeruginosa* biofilms is possible in vitro and in animal experiments. Alginate is a polymer (MW 28,000–1,550,000) consisting of blocks of mannuronic acid and guluronic acid kept together by Ca^++^ which strongly binds to the guluronic blocks. Oligo-guluronic acid (blocks of 12 mainly guluronic acids) has a greater affinity to Ca^++^ and therefore solubilizes alginate in vitro but also in animal experiments (Figure 9) [61]. However, clinical trials with this interesting approach were unfortunately stopped some years ago.

## 5. Conclusions

In vitro experiments, animal experiments and clinical investigations have been carried out to improve treatment of *P. aeruginosa* biofilm infections in CF patients. Gradually, the results of these experiments have improved the prognosis of CF patients with chronic *P. aeruginosa* biofilm lung infections, and the treatment methods used in CF patients have, therefore, spread to other clinical *P. aeruginosa* biofilm infections in, e.g., cilia dyskinesia patients, bronchiectasis patients and patients suffering from chronic wounds [62]. However, antibiotic eradication of such infections has unfortunately not yet been possible and the main reason may be the tolerance of biofilms to antibiotics.

## Figures and Tables

**Figure 1 microorganisms-12-02115-f001:**
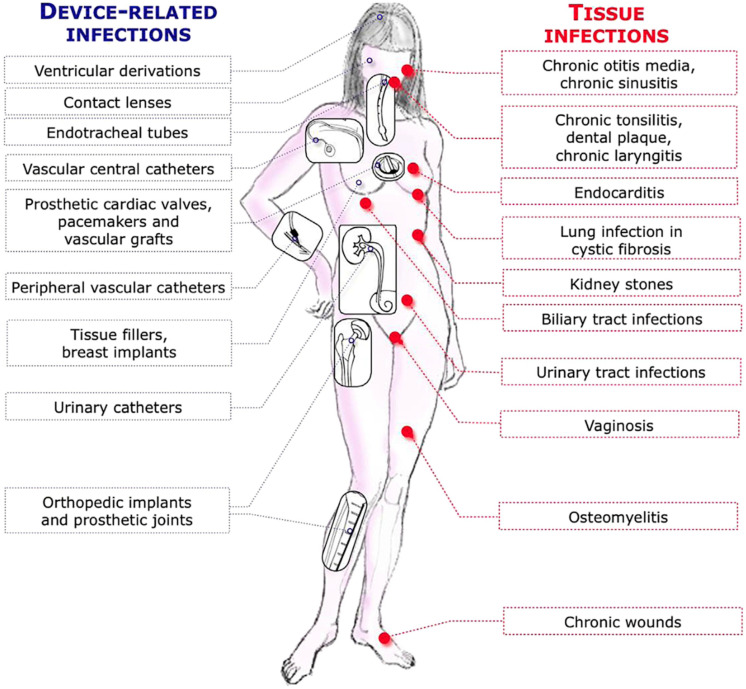
Typical biofilm infections (reproduced with permission from ref. [3], originally).

**Figure 2 microorganisms-12-02115-f002:**
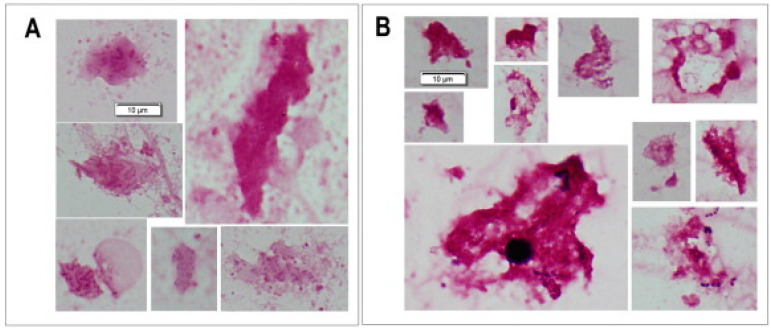
Gram staining of sputum smears from a Danish cystic fibrosis patient CF398 before (**A**) and after (**B**) a 2-week course of suppressive antibiotic therapy administered intravenously and by inhalation. The two bars = 10µm. Gram-negative rods (*P. aeruginosa*) in aggregates (biofilms) embedded in slime from a sputum smear preparation. There was a major decrease in CFU and a major improvement in lung function after therapy. Although biofilms persisted in the sputum, they appeared more condensed (reproduced with permission from ref. [18], Supplementary Material).

**Figure 3 microorganisms-12-02115-f003:**
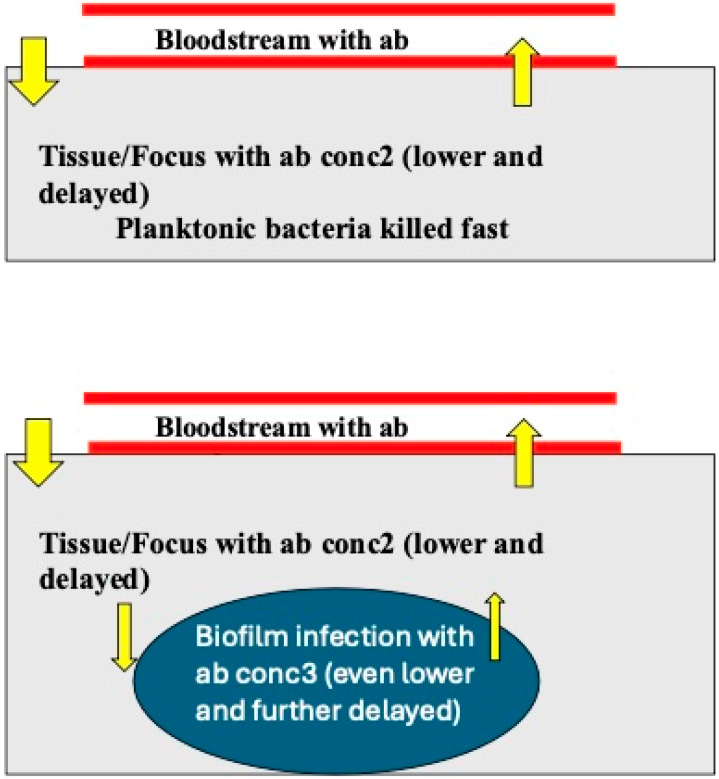
A diagram showing the two general Pharmacokinetic (PK) compartments which characterize antibiotic therapy of acute infections caused by planktonically growing bacteria (the upper part of the diagram) and of the three PK compartments which characterize antibiotic therapy of biofilm infections, (1) the blood stream, (2) the interstitial fluid/tissue and (3) the bacterial biofilm (the lower part of the diagram), with an even lower free concentration (based on ref. [23], but the diagram is not present in the reference (drawn by Claus Moser)).

**Figure 4 microorganisms-12-02115-f004:**
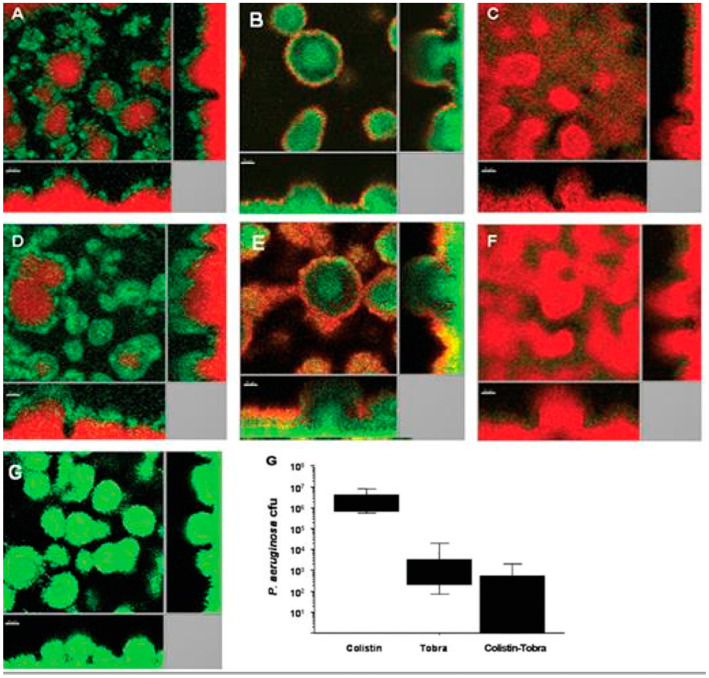
The distribution of dead and live cells in *P. aeruginosa* biofilms in a flow cell visualized by three-dimensional confocal laser scanning microscopy from the top and from two sides. The three-dimensional imaging of the biofilm allowed for the localization of dead (red) and surviving (green) bacteria in the mushroom-like biofilm structures. Biofilms were grown in laminar flow for 4 days at 30 °C and then were continuously exposed to 10 × the minimum inhibitory concentration (MIC) of colistin (**A**,**D**), tobramycin (**B**,**E**) and a combination of both drugs (**C**,**F**) for 24 h (**A**–**C**) and 48 h (**D**–**F**). (**G**) The untreated *P. aeruginosa* biofilm. Live cells appear green as a result of green fluorescent protein expression, and dead cells appear red because of propidium iodine staining (propidium iodine does not penetrate the cell wall of living bacteria). Colistin kills the dormant deep center part of the biofilm, tobramycin kills the metabolizing surface of the biofilm and colistin + tobramycin kills the entire biofilm (synergy). The right lower part of the figure (also named (**G**)): The efficiency of antibiotics against *P. aeruginosa* in a rat lung biofilm infection model. Rats were challenged intratracheally with alginate beads containing 1 × 10^8^ cfu/mL *P. aeruginosa* strain PAO1 (cfu = colony forming units) and then treated with 64× the minimum inhibitory concentration of colistimethate sodium (colistin), tobramycin (tobra) or combinations of these antibiotics. After 7 days, the *P. aeruginosa* cfu were determined and the results for colistin vs. tobra, colistin vs. colistin–tobra, and tobra vs. colistin–tobra were all significantly different, *p* < 0.05. (Reproduced from ref. [42] by permission).

**Figure 5 microorganisms-12-02115-f005:**
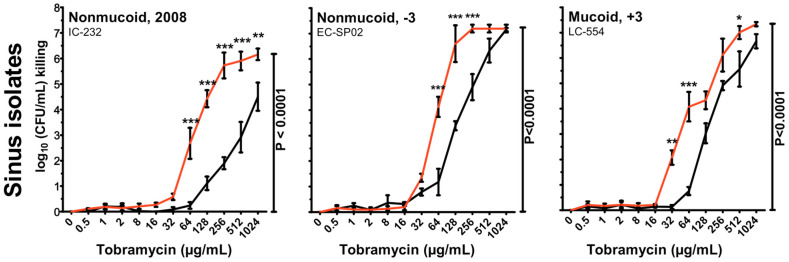
Tobramycin killing of three anoxic aggregating (biofilms) not-related *P. aeruginosa* isolates from sinus paranasales of three different CF patients (IC-232 isolated 2008 (non-mucoid phenotype), EC-SP02 from 2008 (non-mucoid phenotype) taken 3 years before (−3) chronic lung infection started, LC-554 from 2010 (mucoid phenotype) taken 3 years after (+3) the onset of chronic biofilm infection). The aggregates were grown at 37 °C in the agarose-embedded aggregate model for three days during anoxic conditions with NO_3_^−^ as the electron acceptor. The effect of anoxic (black line) or hyperbaric oxygen (90 min. at 100% O_2_ at 2.8 bar, red line) conditions; error bars indicate the mean ± SEM (*n* = 3). Two-way ANOVA with Bonferroni multiple comparison tests; *: *p* ≤ 0.05, **: *p* ≤ 0.01 and ***: *p* ≤ 0.001. (The mean tobramycin concentration in the sputum of cystic fibrosis patients during tobramycin inhalation therapy is >1024 µg/mL [48].) (Reproduced from reference [47] with permission.)

**Figure 6 microorganisms-12-02115-f006:**
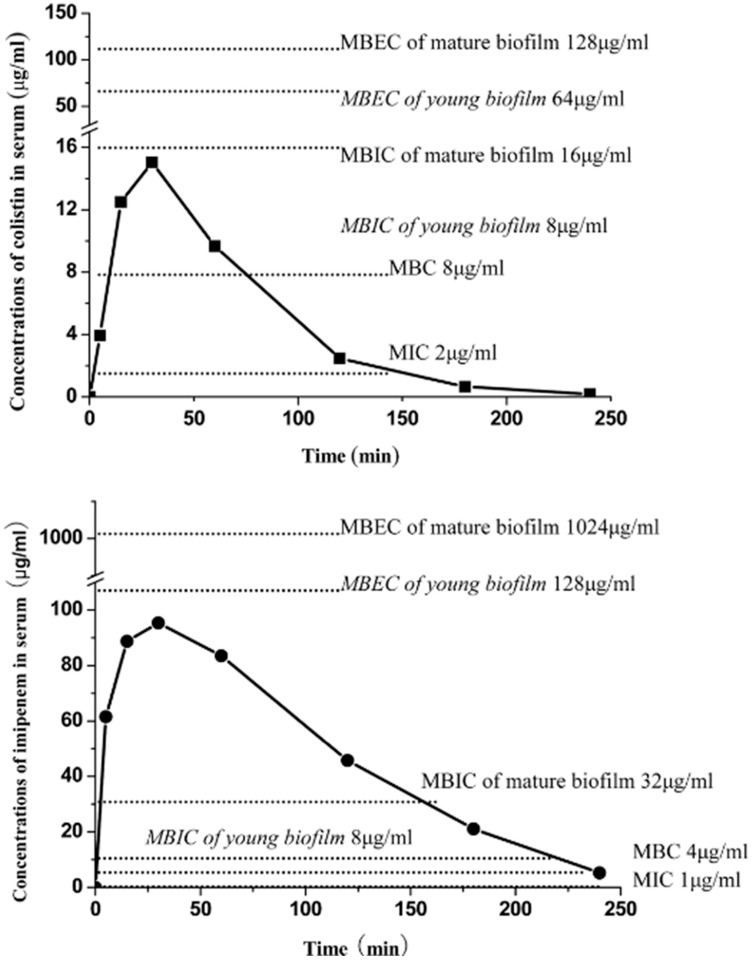
Pharmacokinetics in mouse blood of colistin and imipenem versus minimum inhibitory concentration (MIC), Minimum Bactericidal Concentration (MBC), Minimum Biofilm Inhibitory Concentration (MBIC) and Minimum Biofilm Eradication Concentration (MBEC) of *P. aeruginosa* PAO1. Black squares: 16 mg/kg of colistin; black circles: 64 mg/kg of imipenem with one-dose intraperitoneal administration. Eradication of (MBEC) young (24 h old) or mature biofilms (3 or 7 days old) cannot be achieved. Compare Figure 6 (mice) and Figure 7 (cystic fibrosis patients). (Reproduced from ref. [13] with permission.)

**Figure 7 microorganisms-12-02115-f007:**
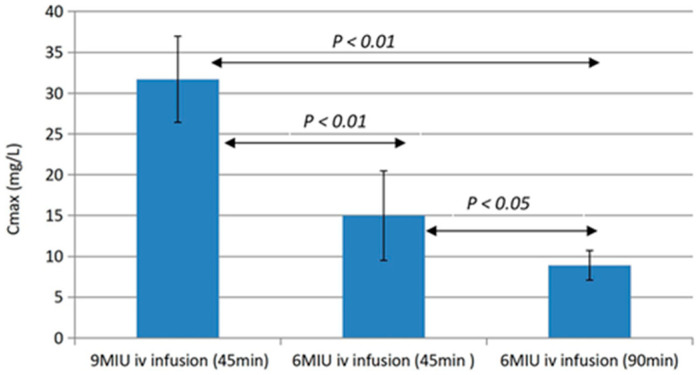
A comparison of the peak concentration (Cmax) of colistin in plasma with IV infusion of one dose of 6 International Million Units (IMIU) and 9 IMIU colistinmethate sodium in 5 cystic fibrosis patients. Compare Figure 6 (mice) and Figure 7 (cystic fibrosis patients). (Reproduced from ref. [49] with permission.)

**Figure 8 microorganisms-12-02115-f008:**
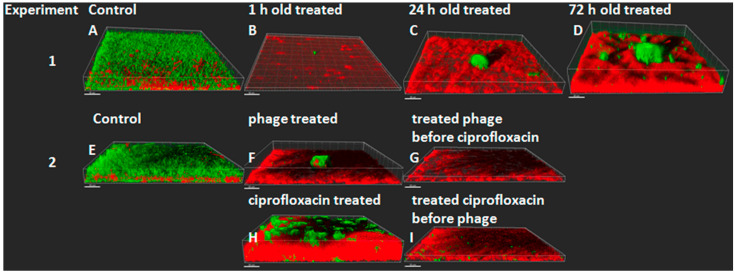
(**1**) The effect of a single phage treatment on *P. aeruginosa* PAO1 biofilms of different ages in flow cells. (**2**) The effect of a combination of phage and ciprofloxacin treatment on PAO1 biofilms. Living cells are green cells (tagged with green fluorescent protein) and dead cells are red (propidium iodide staining). The images show perspective 3D biofilm views. The scale bars are 30 µm. (**1**) (**A**), control; (**B**), phage treatment of a 1 h old biofilm; (**C**), phage treatment of a 24 h old biofilm; (**D**), phage treatment of a 72 h old biofilm. (**2**) (**E**), control; (**F**), phage treatment of a 24 h old biofilm; (**G**), combination treatment of phages followed 5 h later by continuous treatment with ciprofloxacin (0.5 mg/L). (**H**), continuous treatment with ciprofloxacin (0.5 mg/L) of a 24 h old biofilm; (**I**), combination treatment with ciprofloxacin followed 5 h later by the addition of phages. (**1**) Early treatment (1 h biofilm) was most efficient, whereas later treatment gave rise to phage-resistant biofilm colonies (green, (**C**,**D**)). Ciprofloxacin treatment also gave rise to ciprofloxacin-resistant colonies (green, (**H**). Combination treatment with ciprofloxacin and the phage was efficient and prevented the development of resistance independent of the sequence of phage and ciprofloxacin treatment (**G**,**I**). (Treatment with a mixture of 3 different phages or the use of each individual phage gave similar results.) (Reproduced from reference [55] with permission.)

**Figure 9 microorganisms-12-02115-f009:**
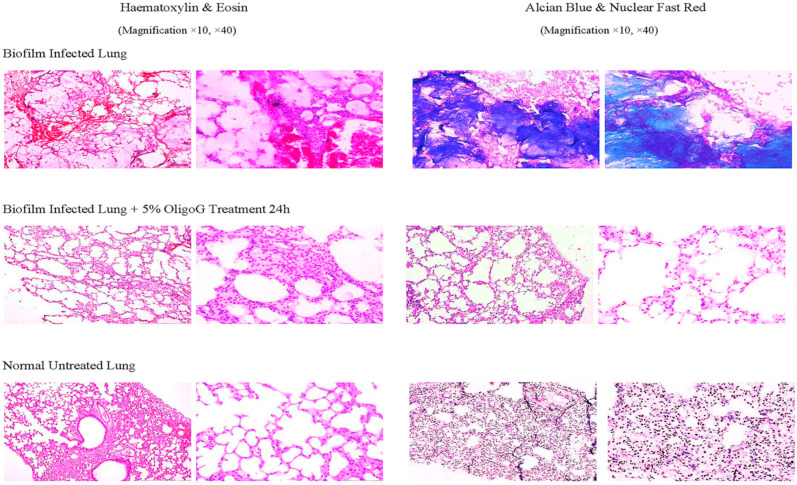
Histopathology of Hematoxylin- and Eosin-stained and Alcian Blue-stained (polysaccharide staining) and Nuclear Fast Red-stained (staining of nuclei) sections of lung tissues from controls and OligoG-treated mice. Mice treated with OligoG exhibited a marked reduction in Alcian blue staining of the alginate, reflecting the significant disruption of biofilms in the infected lungs. (The OligoG used is the OligoG CF5/20 which was used for a clinical trial in cystic fibrosis patients as a solubilizer of sputum.) (Reproduced from ref. [52] with permission.)

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
