# Peer review of "Pseudomonas aeruginosa in the Frontline of the Greatest Challenge of Biofilm Infection—Its Tolerance to Antibiotics"

_microorganisms, 2024, doi:10.3390/microorganisms12112115_

Round 1
Reviewer 1 Report
Comments and Suggestions for Authors
Dear editor and authors
Concerning MS titled" Pseudomonas aeruginosa in the frontline of the greatest challenge of biofilm infection – tolerance to antibiotics.
The MS concerned with the challenges facing treatment of biofilm associated infection, especially tolerance and microbial resistance in the biofilm matrix.
I have some raised points to be addressed
Major
- Fig. 2 is not clear, the resolution is not suitable
- Concerning The title "Tolerance mechanisms of P. aeruginosa"; The section needs to be divided into subtitles "Tolerance to imipenem" "Tolerance to colistin" and so on for more clarification
- Also, phage therapy needs to be separated as subtitle and more examples and data discussion need to be added.
- Line 239; why the best way to study the action of antibiotics
on P. aeruginosa biofilms is confocal laser scanning microscopy, how the adhesion to the microtiter plates is not reliable method.
Minor
Line 6; e.g. aminoglycosides to be such as aminoglycosides.
Same point line 227.
Figures are self-representing so all abbreviations of the figure should be mentioned full in the figure legend such as Fig 7 what is the MIU
Comments on the Quality of English LanguageDear editor and authors
Concerning MS titled" Pseudomonas aeruginosa in the frontline of the greatest challenge of biofilm infection – tolerance to antibiotics.
The MS concerned with the challenges facing treatment of biofilm associated infection, especially tolerance and microbial resistance in the biofilm matrix.
I have some raised points to be addressed
Major
- Fig. 2 is not clear, the resolution is not suitable
- Concerning The title "Tolerance mechanisms of P. aeruginosa"; The section needs to be divided into subtitles "Tolerance to imipenem" "Tolerance to colistin" and so on for more clarification
- Also, phage therapy needs to be separated as subtitle and more examples and data discussion need to be added.
- Line 239; why the best way to study the action of antibiotics
on P. aeruginosa biofilms is confocal laser scanning microscopy, how the adhesion to the microtiter plates is not reliable method.
Minor
Line 6; e.g. aminoglycosides to be such as aminoglycosides.
Same point line 227.
Figures are self-representing so all abbreviations of the figure should be mentioned full in the figure legend such as Fig 7 what is the MIU
Author Response
Fig. 2 is not clear, the resolution is not suitable
Response: A photo with better resolution is now used
- Concerning The title "Tolerance mechanisms of P. aeruginosa"; The section needs to be divided into subtitles "Tolerance to imipenem" "Tolerance to colistin" and so on for more clarification
- Response: Yes we have inserted subtitles
- Also, phage therapy needs to be separated as subtitle and more examples and data discussion need to be added.
- Response: Yes we have expanded the phage-section
- Line 239; why the best way to study the action of antibiotics
on P. aeruginosa biofilms is confocal laser scanning microscopy, how the adhesion to the microtiter plates is not reliable method.
Response: We have explained the reason: Probably the best way to study the action of antibiotics on P. aeruginosa biofilms is confocal laser scanning microscopy (CLSM) of biofilm formation in flow cells, since the action of antibiotics on both the surface and the center of the biofilm can be visualized, but this method is only suitable for research purpose (Figure 4) (41)(42)
Minor
Line 6; e.g. aminoglycosides to be such as aminoglycosides.
Response: done!
Same point line 227.
Response: Done: ...
Figures are self-representing so all abbreviations of the figure should be mentioned full in the figure legend such as Fig 7 what is the MIU
Response: done:...of one-dose 6 International Million Units (IMIU)...
Reviewer 2 Report
Comments and Suggestions for Authors
Comments for Authors:
The review manuscript focuses on the treatment of one of the ESKAPE pathogens, Pseudomonas aeruginosa biofilms that are tolerant to antibiotics and require new effective therapies.
The review focuses mostly on lung and sinus related infections from cystic fibrosis patients and the treatments of these infections. The focus on the lung/mucus/respiratory system in cystic fibrosis patients should reflect in the title. It offers insight into the methodology, however, there are no comparisons between the biofilm formed in the lung and the biofilm in other types of infections (skin, wounds etc.) that are shown in Figure 1. No elaborate details are included to describe the mechanism of the biofilm formation.
Several figures are included that are reproductions from other publications. Most of them could be replaced with a sentence or small table summarizing the relevant information and the citation, or need improvement or should be removed, for example:
1. The gender specific Figure 1 – please remove and replace with a table.
2. Figure 2, which is also a reproduction, has very low resolution and the details of bacteria cannot be distinguished from the smear.
3. Figure 3 is very large, with distracting, strong colors, while the information it carries could be summarized in a sentence.
4. Figure 4, similarly to Figure 2, is a reproduction of a publication with low resolution, and the size bars are missing.
5. Figure 7 and Figure 8 are reproductions of low resolution.
The review is comprehensive in the sense that several aspects of the biofilm formation and treatment are addressed mostly for the lung colonization, however, it could be better organized for easier reading.
However, it could include more sections for example an organ-specific description of the differences if any between the different biofilms (which is missing from the current manuscript), whether they are in vitro aggregates, cells grown in flow cells, animal experiments or clinical tests, the different methods used in their analysis/visualization, and the effect of antibiotic treatment and future directions.
There are staining techniques that are applied in biofilm studies, for example, the crystal violet assay, based on the ability of the dye to stain DNA can provide quantitative information about the relative density of cells, published earlier by:
1. Wilson et al. Quantitative and Qualitative Assessment Methods for Biofilm Growth: A Mini-review. Res Rev J Eng Technol. 2017, 6(4).
2. Bakkiyaraj and Pandian. In vitro and in vivo antibiofilm activity of a coral associated actinomycete against drug resistant Staphylococcus aureus biofilms. Biofouling: The Journal of Bioadhesion and Biofilm Research. 2010, 26, 6, 711-717.
3. Cate et al. Microorganisms 2024, 12(7), 1500; https://doi.org/10.3390/microorganisms12071500
In the “Way Forward” section (Page 8) the Authors state that “Probably the best way to study the action of antibiotics on P. aeruginosa biofilms is confocal laser scanning microscopy (CLSM) of biofilm formation in flow cells, but such methods are only suitable for research purpose”.
While high resolution microscopy is an important and great tool to study biofilm formation, the inclusion of quantitative methods should be essential in this field, in both applied and laboratory studies.
Minor Comment:
Page 8, Lines 262-263: The values p≤0.05, 262 **: p≤0.01, ***: p≤0.01 should be p≤0.05, 262 **: p≤0.01, ***: p≤0.001?
Author Response
1) The review focuses mostly on lung and sinus related infections from cystic fibrosis patients and the treatments of these infections. The focus on the lung/mucus/respiratory system in cystic fibrosis patients should reflect in the title. It offers insight into the methodology, however, there are no comparisons between the biofilm formed in the lung and the biofilm in other types of infections (skin, wounds etc.) that are shown in Figure 1. No elaborate details are included to describe the mechanism of the biofilm formation.
Response: The title has been changed: Pseudomonas aeruginosa in the frontline of the greatest challenge of biofilm infection – tolerance to antibiotics. Review of the experience from cystic fibrosis airway infection.
As the title indicate, there is no reason to include comparison to other biofilm infections. The reader is referred to reference 3) where I and my co-workers have described such infections in detail.
2) Several figures are included that are reproductions from other publications. Most of them could be replaced with a sentence or small table summarizing the relevant information and the citation, or need improvement or should be removed, for example:
- The gender specific Figure 1 – please remove and replace with a table.
- Figure 2, which is also a reproduction, has very low resolution and the details of bacteria cannot be distinguished from the smear.
- Figure 3 is very large, with distracting, strong colors, while the information it carries could be summarized in a sentence.
- Figure 4, similarly to Figure 2, is a reproduction of a publication with low resolution, and the size bars are missing.
- Figure 7 and Figure 8 are reproductions of low resolution.
The review is comprehensive in the sense that several aspects of the biofilm formation and treatment are addressed mostly for the lung colonization, however, it could be better organized for easier reading.
Response: 1) Ice disagree, 2) The same figure with much better resolution is now replacing the original figure 2. 3) We disagree, The idea of a 3rd compartment is new and cannot be replaced by a sentence. 4) We disagree, This is the normal way to show such results and the resolution is fine and size bars are not necessary, since it is the difference between the red (dead) and the green (live) part of the biofilms that is interesting, not the size. We have added a small figure showing the results of rat experiments in the same reference - since our editor asked us to expand our manuscript. 5) We have now used a fig. 7 of better resolution, and concerning fig. 8 we refer to our answer to 4).
3) The review is comprehensive in the sense that several aspects of the biofilm formation and treatment are addressed mostly for the lung colonization, however, it could be better organized for easier reading.
Response: We refer to our response to reviewer 1) where we describe, that we have used a number of subtitles to make the reading easier.
4) However, it could include more sections for example an organ-specific description of the differences if any between the different biofilms (which is missing from the current manuscript), whether they are in vitro aggregates, cells grown in flow cells, animal experiments or clinical tests, the different methods used in their analysis/visualization, and the effect of antibiotic treatment and future directions.
Response: We refer to our response to 1) above.
5) There are staining techniques that are applied in biofilm studies, for example, the crystal violet assay, based on the ability of the dye to stain DNA can provide quantitative information about the relative density of cells, published earlier by:
- Wilson et al. Quantitative and Qualitative Assessment Methods for Biofilm Growth: A Mini-review. Res Rev J Eng Technol. 2017, 6(4).
- Bakkiyaraj and Pandian. In vitro and in vivo antibiofilm activity of a coral associated actinomycete against drug resistant Staphylococcus aureus biofilms. Biofouling: The Journal of Bioadhesion and Biofilm Research. 2010, 26, 6, 711-717.
- Cate et al. Microorganisms 2024, 12(7), 1500; https://doi.org/10.3390/microorganisms12071500
In the “Way Forward” section (Page 8) the Authors state that “Probably the best way to study the action of antibiotics on P. aeruginosa biofilms is confocal laser scanning microscopy (CLSM) of biofilm formation in flow cells, but such methods are only suitable for research purpose”.
While high resolution microscopy is an important and great tool to study biofilm formation, the inclusion of quantitative methods should be essential in this field, in both applied and laboratory studies.
Response: The suggestions of the reviewer has nothing to do with tolerance of biofilms to antibiotics. Our group uses many other methods for biofilm studies including crystal violet staining but that is irrelevant to this survey.
6) Page 8, Lines 262-263: The values p≤0.05, 262 **: p≤0.01, ***: p≤0.01 should be p≤0.05, 262 **: p≤0.01, ***: p≤0.001?
Response: Thank you! We have now corrected the sentence:
Two-way ANOVA with Bonferroni multiple comparison tests, *: p≤0.05, **: p≤0.01, ***: p≤0.001.
Round 2
Reviewer 2 Report
Comments and Suggestions for Authors
This Reviewer would like to thank the Authors for their revisions. However, a few comments remained that should be addressed.
1. Figure 1 is referred to as "Biofilm infections may be found in all anatomical locations of the patients". The figure (1) should be gender neutral; in the current form it suggests that only female anatomy is susceptible to biofilm formation.
2. Figure 2 size bars should be of higher resolution.
3. Figure 3 is too large and too colored. If the Authors insist on including this figure, should at least reformat it.
4. Figure 4 should have numbers on the size bars. Overall, the resolution of the figure is suboptimal.
5. The earlier comment of the Reviewer in the Revision 1 was a helpful suggestion to use staining techniques to quantitate the biofilm after treatment:
"5) Reviewer: There are staining techniques that are applied in biofilm studies, for example, the crystal violet assay, based on the ability of the dye to stain DNA can provide quantitative information about the relative density of cells, published earlier."
This comment was meant to address the Authors' paragraph about the lack of quantitation: “Probably the best way to study the action of antibiotics on P. aeruginosa biofilms is confocal laser scanning microscopy (CLSM) of biofilm formation in flow cells, since the action of antibiotics on both the surface and the center of the biofilm can be visualized, but this method is only suitable for research purpose (Figure 4) (41)(42). At the present time, the way forward is animal experiments based on promising results from CLSM of biofilms exposed for antibiotics and eventually followed by clinical trials (Figure 4) (42).”
The Authors seem to be confused based on their response:
" Authors' Response: The suggestions of the reviewer has (have) nothing to do with tolerance of biofilms to antibiotics. Our group uses many other methods for biofilm studies including crystal violet staining but that is irrelevant to this survey."
Author Response
- Figure 1 is referred to as "Biofilm infections may be found in all anatomical locations of the patients". The figure (1) should be gender neutral; in the current form it suggests that only female anatomy is susceptible to biofilm formation. Response: The figure was originally published in 2014-15 in the ESCMID guideline for diagnosis and treatment of biofilm infections (ref. 3) and can therefore not be changed. Furthermore, the figure shows that biofilm infections occur on tissue fillers and breast implants and as vaginosis which are female infections as indicated on the figure.
- Figure 2 size bars should be of higher resolution. Response: The size bars on the figure (the new edition of fig. 2 with better resolution) is completely clear. And again, this figure was originally published in 2017 (ref. 18) and can therefore not be changed.
- Figure 3 is too large and too colored. If the Authors insist on including this figure, should at least reformat it. Response: We insist on including the figure, but the journal may print it as a small figure.
- Figure 4 should have numbers on the size bars. Overall, the resolution of the figure is suboptimal. Response: This figure was originally published in 2010 (ref. 42) and accepted by Journal of Infectious Diseases in the present form and can therefore not be changed. The resolution is NOT suboptimal and every aspect of the interaction of colistin, tobramycin and the combination can be easily seen like in the original publication. There was no numbers on the size bars in the original publication (ref. 42) and neither in the text or in Materials and Methods in ref. 42, because they are not necessary for understanding of the figure.
-
5. The earlier comment of the Reviewer in the Revision 1 was a helpful suggestion to use staining techniques to quantitate the biofilm after treatment:
"5) Reviewer: There are staining techniques that are applied in biofilm studies, for example, the crystal violet assay, based on the ability of the dye to stain DNA can provide quantitative information about the relative density of cells, published earlier."
This comment was meant to address the Authors' paragraph about the lack of quantitation: “Probably the best way to study the action of antibiotics on P. aeruginosa biofilms is confocal laser scanning microscopy (CLSM) of biofilm formation in flow cells, since the action of antibiotics on both the surface and the center of the biofilm can be visualized, but this method is only suitable for research purpose (Figure 4) (41)(42). At the present time, the way forward is animal experiments based on promising results from CLSM of biofilms exposed for antibiotics and eventually followed by clinical trials (Figure 4) (42).”
The Authors seem to be confused based on their response:
" Authors' Response: The suggestions of the reviewer has (have) nothing to do with tolerance of biofilms to antibiotics. Our group uses many other methods for biofilm studies including crystal violet staining but that is irrelevant to this survey."
Response: We refer to our response above and can assure, that we are not confused.